# Dysregulation of the NLRP3 Inflammasome and Promotion of Disease by IL-1β in a Murine Model of Sandhoff Disease

**DOI:** 10.3390/cells14010035

**Published:** 2025-01-01

**Authors:** Nick Platt, Dawn Shepherd, David A. Smith, Claire Smith, Kerri-Lee Wallom, Raashid Luqmani, Grant C. Churchill, Antony Galione, Frances M. Platt

**Affiliations:** 1Department of Pharmacology, University of Oxford, Mansfield Road, Oxford OX1 3QT, UK; nick.platt@pharm.ox.ac.uk (N.P.); dawn.shepherd@pharm.ox.ac.uk (D.S.); dave.smith@pharm.ox.ac.uk (D.A.S.); claire.smith@pharm.ox.ac.uk (C.S.); kerri-lee.wallom@pharm.ox.ac.uk (K.-L.W.); grant.churchill@pharm.ox.ac.uk (G.C.C.); antony.galione@pharm.ox.ac.uk (A.G.); 2Nuffield Department of Orthopaedics, Rheumatology and Musculoskeletal Sciences, University of Oxford, Windmill Road, Oxford OX3 7LD, UK; raashid.luqmani@ouh.nhs.uk

**Keywords:** Sandhoff disease, GM2 gangliosidosis, lysosomal storage disease, NLRP3 inflammasome, IL-1β, inflammation

## Abstract

Sandhoff disease (SD) is a progressive neurodegenerative lysosomal storage disorder characterized by GM2 ganglioside accumulation as a result of mutations in the *HEXB* gene, which encodes the β-subunit of the enzyme β-hexosaminidase. Lysosomal storage of GM2 triggers inflammation in the CNS and periphery. The NLRP3 inflammasome is an important coordinator of pro-inflammatory responses, and we have investigated its regulation in murine SD. The NLRP3 inflammasome requires two signals, lipopolysaccharide (LPS) and ATP, to prime and activate the complex, respectively, leading to IL-1β secretion. Peritoneal, but not bone-marrow-derived, macrophages from symptomatic SD mice, but not those from pre-symptomatic animals, secrete the cytokine following priming with LPS with no requirement for activation with ATP, suggesting that such NLRP3 deregulation is related to the extent of glycosphingolipid storage. Dysregulated production of IL-1β was dependent upon caspase activity but not cathepsin B. We investigated the role of IL-1β in SD pathology using two approaches: the creation of *hexb*^−/−^*Il1r1*^−/−^ double knockout mice or by treating *hexb*^−/−^ animals with anakinra, a recombinant form of the IL-1 receptor antagonist, IL-1Ra. Both resulted in modest but significant extensions in lifespan and improvement of neurological function. These data demonstrate that IL-1β actively participates in the disease process and provides proof-of-principle that blockade of the pro-inflammatory cytokine IL-1β may provide benefits to patients.

## 1. Introduction

Sandhoff disease (SD), a GM2 gangliosidosis disorder, is a rare, autosomal recessive, prematurely fatal lysosomal storage disease (LSD) caused by mutations in *HEXB*, which encodes the β subunit of β-hexosaminidase [1,2]. Loss of this enzyme activity results primarily in the lysosomal accumulation of GM2 ganglioside in the CNS and periphery, which triggers progressive neurodegeneration [3,4]. Clinical symptoms include developmental regression, dystonia, dysphagia and ataxia [1,5]. Similar to other LSDs, the extent of residual enzyme activity significantly impacts upon the age of disease onset and the rate of disease progression. Although there is a continuum of clinical presentations, discrete infantile, juvenile and late-onset forms of the disease have been characterized [1].

Studies of the underlying pathogenic mechanisms in SD and pre-clinical testing of potential therapeutics have benefited greatly from the availability of authentic animal models of SD [6]. Mice lacking β-hexosaminidase activity (*hexb*^−/−^, SD mice) store the gangliosides GM2 and GA2 in the CNS and undergo rapid progressive neurodegeneration, with evidence of ataxia at ~11 weeks of age, motor dysfunction from ~12 weeks of age, and death at ~15 weeks of age [7,8]. Studies of *hexb*^−/−^ mice have highlighted the occurrence of neuroinflammation and immune cell activation in the CNS that parallels patient studies [9]. Over the course of CNS disease in the *hexb*^−/−^ mouse, there is significant upregulation of multiple indicators of neuroinflammation, including enhanced MHC class II and CD68 expression on activated microglia and the production of pro-inflammatory cytokines [10]. Evidence that progressive CNS inflammation correlates with the onset of clinical signs [10], that substrate reduction therapy (SRT), which reduces glycolipid biosynthesis, significantly delays the appearance of inflammation while extending life span [8], and that a combination of non-steroidal anti-inflammatory drugs with substrate reduction therapy provides greater survival [11] supports the premise that inflammation actively contributes to pathogenesis in the GM2 gangliosidoses.

Interleukin-1 beta (IL-1β) is a prototypic proinflammatory cytokine that evokes fever, hypotension, and release of acute phase proteins and orchestrates multiple aspects of inflammatory and immune responses [12,13,14]. IL-1β not only stimulates activities of innate immune cells, such as promoting recruitment of inflammatory cell populations, the secretion of other cytokines, and enhancing cell survival and effector mechanisms, but also impacts adaptive immune cells, particularly through activation and differentiation of naïve T cells [15]. Bioactive IL-1β is generated via the activation of intracellular protein complexes termed inflammasomes. The best characterized is the nucleotide-binding oligomerization, leucine-rich repeat and pyrin domain-containing protein 3 inflammasome (NLRP3) that consists of the sensor protein NLRP3, the adaptor caspase-recruitment domain protein, ASC, and the effector caspase-1 [16] (see Figure 1a).

The canonical generation of mature IL-1β by the NLRP3 inflammasome entails two steps: initial priming resulting from the binding of cytokines, such as TNFα or pathogen-associated molecular patterns (PAMPs) to cognate receptors, resulting in the synthesis of ProIL-1β and stabilization of NLRP3 that serves as the scaffold required for inflammasome function (step 1); and step 2, which involves the subsequent inflammasome activation by numerous stimuli such as extracellular ATP or particulates and crystals, triggering the formation of the inflammasome protein complex and the auto-activation of caspase-1, which cleaves ProIL-1β to allow secretion of the bioactive cytokine (Figure 1a) [16,17,18]. This two-step regulation of the inflammasome has evolved to prevent the inappropriate triggering of chronic inflammation. The importance of this precise regulation of IL-1β production is underscored by the clinical severity of a group of inherited, dominant auto-inflammatory disorders, including cryopyrin-associated periodic syndromes such as Muckle–Wells that are caused by mutations in NLRP3 that enhance inflammasome activation and increase processing of the cytokine precursor, leading to chronic dysregulated inflammation [19].

In light of the published evidence of enhanced levels of IL-1β in the brain concurrent with disease progression in *hexb*^−/−^ mice [10], we were interested in exploring the extent to which the cytokine affects the underlying disease-promoting mechanisms. Here, we report unexpected dysregulation of the NLRP3 inflammasome in *hexb*^−/−^ murine macrophages (Mϕ) that coincides with the development of neurological signs in SD mice. We show that genetic and biologic inhibition of IL-1β extends the lifespan of SD mice and improves neuromuscular function. These data demonstrate a key role of IL-1β in the pathophysiology of the disease and its potential as a therapeutic target for clinical intervention.

## 2. Material and Methods

### 2.1. Animals

Sandhoff disease mice (*hexb*^−/−^) [7] on a C57Bl/6 genetic background were kept at the University of Oxford under specific pathogen-free conditions and maintained through heterozygote breeding to generate *hexb*^−/−^ and control genotypes.

*Il1r1*^−/−^ mice were a generous gift from Professor Vincenzo Cerundolo (Weatherall Institute, University of Oxford). Doubly deficient animals (*hexb*^−/−^*Il1r1*^−/−^) were generated by crossing the two strains followed by intercrossing of the offspring. All mice used were identified and genotyped by PCR of DNA prepared from ear notches. Primer details: *HEXB* genotyping primers: AATTTAAAATTCAGGCCTCGA, CATAGCGTTGGCTACCCGTGA, CATTCTGCAGCGGTGCACGGC; *Il1r1* primers, wild-type allele type, GGTTTGAATGTTGGGGTTTG, CACCACCACCTGGCTACTTT; null allele, TCTGGACGAAGAGCATCAGGG, CAAGCAGGCATCGCCATG. A humane end-point was applied in studies of survival and was defined to be when mice became moribund and were no longer able to right themselves within 30 s of being laid on their side. All animal use was approved under the authority of a license issued by the UK Home Office (Animals [Scientific procedures] Act 1986).

### 2.2. Mouse Treatments and Cell Recovery

*hexb*^−/−^ mice were injected intraperitoneally with anakinra ((Kineret^®^) (Swedish Orphan Biovitrum, Stockholm, Sweden), at a dose of 1 mg/kg, three times weekly from three weeks of age.

### 2.3. Cells

Resident peritoneal macrophages (RPMϕ) were recovered by peritoneal lavage, centrifuged, and plated in complete medium (RPMI 1640 (Sigma, Dorset, UK) containing 10% fetal calf serum (*v*/*v*), 1% glutamine and 1% penicillin–streptomycin) (all Sigma, Dorset, UK). Cells were allowed to recover overnight before experimentation, and non-adherent cells were washed off so as to significantly enrich for RPMϕ. Bone-marrow-derived macrophages (BMMϕ) were generated as described [20] and cultured in complete medium supplemented with 20% *v*/*v* L929 cell-conditioned supernatant and harvested on day 6.

### 2.4. LysoTracker^TM^ Staining and Analysis by Flow Cytometry

Live cells were blocked with Fc block (BD Biosciences, Wokingham, Berkshire, UK) prior to staining with titrated anti-F4/80-APC antibody (Biolegend, London, UK) on ice, washed, and then stained with LysoTracker^TM^ Green DND-26 (L-7526, ThermoFisher Scientific, Paisley, UK) according to Vruchte et al. [21]. Events were immediately collected on a FACSCanto II cytometer (BD Biosciences, Wokingham, Berkshire, UK). A minimum of 1 × 10^4^ events were accumulated for each sample. Data were gated and analyzed using FlowJo software (FlowJo, LLC, version 10, Ashland, OR, USA).

### 2.5. Cell Stimulations, Immunoblotting, and Specific ELISA

Plated cells were either primed with ultrapure lipopolysaccharide (tlrl-eblps, Invivogen, Toulouse, France) at 50 ng/mL for 6 h or peptidoglycan (tlrl-pgnek, Invivogen) at 10 μg/mL for 6 h and where indicated, activated with 5 mM ATP (Sigma, Dorset UK) for 1 h, and culture supernatants were collected. For experiments involving inhibitors, 10 μg/mL z-VAD-fmk (tlrl-vad, Invivogen, Toulouse, France) or 5 μM CA-074-Me (C5857, Sigma, Dorset, UK) was added 30 min prior to priming. In the case of uric-acid-mediated activation, primed cells were incubated with 100 μg/mL uric acid crystals (tlrl-msu, Invivogen, Toulouse, France) for 6 h. For immunoblotting, supernatants were precipitated with 1 volume of methanol and 0.25 volumes of chloroform and protein pellets solubilized in sample buffer (Cell Signaling Technology, Leiden, The Netherlands). Samples were separated on 15% polyacrylamide gels and transferred to polyvinylidene membranes (GE Healthcare, Chalfont St Giles, Buckinghamshire, UK) on a Bio-rad Turbo Blot system. Membranes were blocked in tris-buffered saline containing 5% non-dairy fat dried milk powder and then probed with goat anti-IL-1β antibody (AF-401, R&D Systems, Abingdon, Oxfordshire, UK) or rabbit anticaspase-1 antibody (sc-514), Santa Cruz Biotechnology, Heidelberg, Germany), washed, and incubated with horseradish-peroxidase-conjugated donkey antigoat IgG (705-035-147, Jackson Immunoresearch, Ely, Cambridgeshire, UK) or horseradish-peroxidase-conjugated donkey antirabbit IgG (711-035-152). Membranes were washed and developed with SuperSignal^TM^ West Pico Plus Chemiluminescent substrate (Thermo Scientific). Gel images were collected on a ChemiDoc XRS+ Imaging System (Bio-rad, Oxford, Oxfordshire, UK), and images were processed using Imagelab Version 6.1 software (Bio-rad, Oxford, Oxfordshire, UK). Cytokine concentrations in culture supernatants were determined using specific ELISA (IL-1β, BMS6002, Thermo Fisher Scientific, Paisley, UK) according to instructions provided by the manufacturer.

### 2.6. Behavioral Analyses

*hexb*^−/−^ mice display progressive neurodegeneration, which can be quantitatively assessed by evaluating their performance in behavioral tests. Functional tests were performed in a blinded manner. Horizontal bar-crossing was used to evaluate motor coordination and hind limb strength, as described previously [8]. In brief, animals were allowed to grasp a metal bar suspended horizontally between two wooden supports over a cushioned surface using their forepaws only. Latency to cross (climb off onto one of the two supports) or fall from the bar onto the cushioned surface was scored, with a 180 s maximum to termination of the test with a score ranging from +180 to −180 recorded for each animal. If the crossing time was greater than 0, then the score = 180—crossing time; if a falling time was greater than 0, then the score = falling time—180.

Following room acclimatization for a period of 10–30 min, spontaneous activity was measured by placing the mouse in an open field of 540 cm^2^, and rearing events (number of times the mouse reared on its hind legs without the support of the cage wall) were recorded manually for 5 min.

Tremor was measured with a commercial tremor monitor (San Diego Instruments, Ormskirk, Lancashire, UK) according to the manufacturer’s instructions. The monitor was housed on an antivibration table, and mice were analyzed weekly for 256 s, following a period of 30 s acclimatization. Data were collected on a computer via a National Instruments PCI card, and output was analyzed (fast Fourier transform) using LabVIEW 2020 software (National Instruments, Reading, Berkshire, UK) to give a measure of amplitude at each frequency (0–64 Hz).

### 2.7. Statistical Analysis

All statistical analyses were carried out with GraphPad Prism Software (Version 10). Survival curves were constructed using the log-rant test as described previously [11]. Bar-crossing data were analyzed by fitting a four-parameter logistic function to the data [11]. Student’s *t*-test was used to determine the statistical significance of all unpaired observations. Multiple comparisons were analyzed by one-way analysis of variance (ANOVA) followed by post hoc Bonferroni’s correction. Statistical significance was determined compared with controls with *p* < 0.05 as the threshold for significance.

## 3. Results

### 3.1. hexb^−/−^ Resident Peritoneal Macrophages (RPMϕ) from 14-Week-Old Mice Have Enhanced LysoTracker^TM^ Staining


A characteristic of LSDs is an expansion of the acidic lysosomal compartment, which increases with disease progression [21]. A measure of the relative volume of the acidic compartment can be obtained from the fluorescence intensity of live cells stained with the LysoTracker^TM^ probe and analyzed by flow cytometry (FACS) (Appendix A). FACS analysis showed that F4/80^hi^ RPMϕ isolated from symptomatic fourteen-week-old *hexb*^−/−^ mice displayed significantly greater LysoTracker^TM^ Green DND-26 staining than age-matched WT (*hexb^+/+^*) littermates as predicted (Figure 1b).

### 3.2. hexb^−/−^ RPMϕ from Symptomatic SD Mice Display Aberrant NLRP3 Inflammasome Activation and Secretion of IL-1β


To investigate NLRP3-dependent generation of IL-1β, we isolated RPMϕ from late-stage (14-week-old) *hexb*^−/−^ mice and age-matched *hexb*^+/+^ littermates and probed Western blots of proteins from culture supernatants of cells primed with LPS or peptidoglycan (PNG) only (signal 1) or primed and then activated with ATP (signal 1 and 2) (Figure 2a). As expected, we were able to detect secretion of the bioactive cytokine (IL-1β, molecular mass 17 kDa) only in *hexb*^+/+^ RPMϕ cultures that had been both primed and activated (Figure 2a). In contrast, we detected significant amounts of IL-1β in culture supernatants of RPMϕ isolated from 14-week-old SD mice that had only been primed with LPS or PGN (Figure 2a). We also measured the secretion of bioactive caspase-1, which is also released under these conditions and observed results comparable to those for IL-1β; mature caspase-1 (molecular mass ~10 kDa) was only present at significant levels in supernatants of *hexb*^+/+^ RPMϕ that had been primed and activated (signal 1 and signal 2), whereas, for *hexb*^−/−^ RPMϕ isolated from 14-week-old animals, we detected significant secretion by LPS-primed cells (Figure 2b) in the absence of signal 2.

In order to quantify the secretion of IL-1β, we measured the cytokine content of culture supernatants using a specific ELISA (Figure 3). These data confirmed statistically significant secretion by LPS-primed *hexb*^−/−^ RPMϕ from 14-week-old animals but not from primed wild-type cells, but there was comparable production when cells of the two genotypes were primed and activated (signal 1 and signal 2).

### 3.3. Bone-Marrow-Derived Macrophages (BMMϕ) from hexb^−/−^ Mice Require NLRP3 Priming and Activation (Signal 1 and Signal 2) to Produce IL-1β

We were interested in determining whether loss of β-hexosaminidase activity in other Mϕ populations also resulted in the secretion of bioactive cytokine following the single stimulus of NLRP3 priming (signal 1). We generated BMMϕ from 14-week-old *hexb*^−/−^ and age-matched control animals using a standard protocol and harvested cells at day 6. We were unable to measure a significant difference in the intensity of LysoTracker^TM^ staining between the two genotypes (Figure 4a). BMMϕ were primed (signal 1) or primed and activated (signal 1 and signal 2), and IL-1β secretion was measured by ELISA. Priming (signal 1) with either LPS or PNG failed to produce detectable secretion of the cytokine, whereas significant levels were produced after priming and ATP-mediated activation (signal 1 and signal 2) in both genotypes, consistent with canonical NLRP3-mediated generation of IL-1β (Figure 4b).

### 3.4. IL-1β Is Secreted by LPS-Primed RPMϕ Isolated from Symptomatic 12-Week-Old and 14-Week-Old hexb^−/−^ Mice but Not by RPMϕ from Non-Symptomatic 8-Week-Old hexb^−/−^ Mice

We interpreted the failure of a priming stimulus alone to cause secretion of IL-1β by *hexb*^−/−^ BMMϕ to indicate that loss of *HEXB* activity alone is insufficient for aberrant NLRP3 activation. LSDs, including SD, progressively accumulate biochemical substrates that cannot be catabolized [1,2]. We therefore investigated the relationship between aberrant IL-1β secretion by primed RPMϕ and the extent of disease progression in *hexb*^−/−^ mice. RPMϕ were isolated from symptomatic (12-week-old) and late-stage (do not delete ‘late’) symptomatic (14-week-old) *hexb*^−/−^ animals, ages at which multiple features of disease, including quantitative changes in behavior that reflect progressive neurodegeneration together with CNS inflammation [8,10], are displayed. We also studied RPMϕ from 8-week-old pre-symptomatic animals that did not show any of these functional deficits. RPMϕ obtained from age-matched *hexb*^+/+^ mice served as controls. Cells were either untreated, primed with LPS (signal 1), or primed and then activated with ATP (signal 1 and signal 2), and the level of IL-1β secretion was measured by ELISA. *hexb*^+/+^ and *hexb*^−/−^ cells from 8-week-old mice only produced cytokine after LPS priming and ATP-mediated activation (signal 1 and signal 2) (Figure 5a), whereas 12-week-old and 14-week-old *hexb*^−/−^ RPMϕ but not WT cells from animals of the same age secreted significant levels of IL-1β following priming with LPS (signal 1) (Figure 5b,c). These data (summarized in Table 1) indicate a correlation between the occurrence of aberrant NLRP3 activation and the progression of disease in the murine SD model.

### 3.5. Aberrant IL-1β Production by hexb^−/−^ RPMϕ Isolated from Symptomatic Animals Is Prevented by the Caspase-1 Inhibitor, Z-Val-Ala-DL-Asp-fluoromethylketone (zVAD-fmk) but Not by Inhibition of Cathepsin B

NLRP3 activation results in the generation of caspase-1, which catalyzes the cleavage of proIL-1β into its mature form that is secreted and functional [22]. However, NLRP3 activation as a result of lysosomal rupture or destabilization resulting from the uptake of particulates such as silica crystals and aluminum salts is at least in part dependent upon cathepsin B activity, which is blocked by the inhibitor CA-074-Me [23]. To investigate whether cathepsin B activity might be responsible for aberrant NLRP3 activation in *hexb*^−/−^ RPMϕ, we compared the effects of z-VAD-fmk, which blocks caspase-1 activity, and CA-074-Me, which blocks cathepsin B. To verify the specific inhibitory activities of the drugs, we included cells that were primed with LPS and then incubated with monosodium urate (MSU) to cause inflammasome activation via lysosome rupture. Concentrations of IL-1β in culture supernatants were determined by specific ELISA. As expected, we were able to detect significant production of IL-1β by late-stage *hexb*^−/−^ RPMϕ primed with LPS but not by similarly treated *hexb*^+/+^ cells isolated from animals of the same age (Figure 6). z-VAD-fmk significantly reduced cytokine generation by primed and ATP-activated wild-type RPMϕ and similarly, CA-074-Me reduced IL-1β secretion by *hexb*^+/+^ RPMϕ following priming and MSU-mediated activation. While z-VAD-fmk was able to significantly limit production by *hexb*^−/−^ RPMϕ primed with LPS, inhibition of cathepsin B in the same cells had no significant effect. These data suggest that aberrant generation of mature cytokine by *hexb*-deficient cells is dependent upon caspases and is not due to inadvertent cathepsin B activity.

### 3.6. Genetic Deletion of IL-1β Activity Increases the Lifespan of hexb^−/−^ Mice and Reduces Tremors

It has been shown that in mouse models of different LSDs, inflammation actively contributes to disease progression. For example, the beneficial effects of non-steroid anti-inflammatory drugs have been demonstrated in *hexb*^−/−^ mice [11]. In order to evaluate the effects of IL-1β on disease outcome in SD mice, we generated *hexb*^−/−^ mice lacking *interleukin 1 receptor1* (*hexb*^−/−^*Il1r1*^−/−^) and examined whether there was an improvement in symptoms and enhanced survival as a result. *hexb*^−/−^*Il1r1*^−/−^ mice had a modest but significantly increased survival (Figure 7a) and lifespan (Figure 7b) in comparison with control animals, and this was seen with both sexes. The lifespan of *hexb*^−/−^*Il1r1*^−/−^ male mice was 7.3% greater than controls and 8.5% longer for females of the same genotype. We undertook a series of behavioral analyses to determine if any functional improvement had been achieved. Importantly, there was a significant reduction in the amplitude of tremors in doubly deficient mice in comparison with controls (Figure 7c), particularly in low-frequency tremors (~20 Hz) (Figure 7d). However, there was no significant difference between body weights of *hexb*^−/−^*Il1r1*^+/−^ and *hexb*^−/−^*Il1r1*^−/−^ animals, nor was there a measurable improvement in bar crossing, an indicator of relative hind limb functional impairment by *hexb*^−/−^
*Il1r1*^−/−^ mice (Appendix A).

### 3.7. Competitive Inhibition of IL-1β Activity with Interleukin One Receptor Antagonist (IL1Ra) Provides Benefit in hexb^−/−^ Mice

The endogenous regulation of IL-1β bioactivity includes the secretion of naturally occurring IL1Ra, a molecule that competitively inhibits IL-1β binding to its receptor but fails to engage downstream intracellular signal transduction mechanisms, thereby modulating the generation of IL-1-induced inflammatory mediators [12]. A recombinant form of ILRa, anakinra (Kineret^®^), has been used therapeutically in a number of clinical situations [24]. We therefore undertook a pilot study and examined whether *hexb*^−/−^ mice that had been administered anakira showed improvement in survival and/or neurological function in a pilot investigation.

Anakinra-treated *hexb*^−/−^ mice survived significantly longer than controls (Figure 8a), with a mean lifespan of 118.6 days for treated animals as against 111.4 days for vehicle-injected mice, equivalent to a 6.5% increase in survival (Figure 8b). The evaluation of center-rearing behavior, measured as the frequency that animals were able to stand on their hind legs in an open field within a five-minute period, was determined across the lifespan of both cohorts of mice. From 6 weeks of age, there was a trend for anakinra-treated mice to exhibit a greater number of rearing events, which reached statistical significance at 6 and 10 weeks of age, and this rearing activity persisted through to 14 weeks, whereas no rearing was recorded for vehicle-treated mice beyond 10 weeks of age (Figure 8c).

## 4. Discussion

Sphingolipidoses are recessive, prematurely fatal, neurodegenerative disorders triggered by the progressive accumulation of sphingolipids in the lysosome [25,26]. Incomplete catabolism of GM2 ganglioside is characteristic of both Tay–Sachs and SD (which have mutations in alpha and beta subunits of β-hexosaminidase, respectively) [1,27]. Currently, there are no approved treatments for SD or Tay–Sachs disease.

Chronic inflammation resulting from inappropriate activation of the innate immune system is a widespread feature of LSDs, including the GM2 gangliosidoses [1]. It is particularly prominent in those conditions with CNS involvement [28] but is also present in disorders in which disease manifestations are restricted to the periphery [29]. Although it is an almost universal characteristic of LSDs, a mechanistic understanding of how loss of lysosome homeostasis triggers inflammation remains to be defined. Here, in a study of a specific component of the inflammatory response in an authentic murine model of SD, we found dysregulation of the NLRP3 inflammasome, a cytosolic protein complex that is responsible for the generation of IL-1β [16] resulting in chronic inflammation. In a proof-of-principle study, we provide evidence that therapeutic targeting of this pro-inflammatory cytokine has the potential to provide some benefit to patients.

The generation of an authentic mouse model of SD [7] has not only advanced our understanding of the pathophysiology of this disease but, importantly, has facilitated the identification of new points for potential clinical intervention and has expediated pre-clinical testing of novel therapeutic approaches—for example, evaluation of substrate reduction therapy [8], stem cell transplantation [30], and acetyl-DL-leucine [31]. In addition, acetyl-L-leucine has shown efficacy in a clinical Phase 2b trial [32]. Detailed studies of murine models of the GM2 gangliosidoses have provided evidence of shared neuro-inflammatory and neuro-degenerative processes that include microglial activation, neuronal apoptosis, astrogliosis, loss of blood–brain barrier integrity, recruitment of inflammatory cell populations to the brain, and the production of specific pro-inflammatory cytokines and chemokines, [10,33,34,35], all of which correlate with disease progression.

In this study, we focused on IL-1β, a prototypic pro-inflammatory cytokine whose levels in the CNS increase significantly during the later stages of the murine disease [10]. Unexpectedly, in vitro stimulation of RPMϕ isolated from symptomatic *hexb*^−/−^ mice revealed dysregulation of NLRP3 inflammasome activity; priming alone was sufficient to trigger the secretion of the cytokine without the requirement for a second signal to trigger inflammasome activation. Canonical production of IL-1β by the NLRP3 inflammasome is a tightly regulated process so as to avoid inadvertent release of cytokine leading to chronic inflammation [18]. Generation of the bioactive cytokine requires two signals: a priming step (signal 1) that results from receptor-dependent recognition of pathogen-associated molecular patterns (PAMPs), damage-associated molecular patterns (DAMPs), or cytokines that results in transcription and synthesis of the NLRP3 inflammasome components, procaspase 1 and proIL-1β; a second activating signal (signal 2), which in the case of NLRP3 inflammasome, includes a diverse range of stimuli that trigger formation of the inflammasome complex, caspase activation, enzymatic cleavage, and secretion of active cytokine [18]. Interestingly, aberrant generation of IL-1β as a consequence of the perturbation of NLRP3 regulation is observed in other rare monogenic diseases that are members of a group collectively known as autoinflammatory disorders [19]. Mutations in the *NLRP3* gene are responsible for Muckle–Wells syndrome (MWS), chronic infantile neurologic cutaneous syndrome (CINCA), and cryopyrin-associated periodic syndromes (CAPS) [19,36], which present with fevers, aseptic meningitis, arthritis, and cutaneous inflammation of variable severity. Monocytes from MWS patients display enhanced pro-IL-1β processing activity and spontaneously secrete active IL-1β [22]. In MWS, IL-1β is secreted even in the absence of stimulation with LPS, which we did not observe in our study, suggesting that the molecular mechanism causing dysregulation of the NLRP3 inflammasome in MWS is unlikely to be responsible for the novel phenotype we describe in the murine SD model. Because we used Western blotting and specific ELISA to analyze cytokine production, we cannot absolutely exclude the possibility that IL-1β was also secreted by cells other than RPMϕ. However, this seems highly unlikely because we enriched significantly for RPMϕ by adherence and did not detect any difference in the number of cells recovered by peritoneal lavage.

The occurrence of NLRP3 inflammasome dysregulation was age-dependent: RPMϕ from symptomatic and late-stage symptomatic *hexb*^−/−^ mice of 12 and 14 weeks of age which exhibit multiple disease symptoms including tremor, ataxic gate, muscle weakness, and motor dysfunction [30] aberrantly secreted IL-1β in response to priming alone, whereas Mϕ prepared from 8-week-old pre-symptomatic mice displayed canonical, two-signal-dependent IL-1β production. These data suggest a positive relationship between the extent of GM2/GA2 storage and the occurrence of aberrant inflammasome regulation. Furthermore, *hexb*^−/−^ BMMϕ which, unlike *hexb*^−/−^ RMPϕ did not display enhanced LysoTracker^TM^ staining intensity, indicating a lack of significant storage in the acidic compartment, also showed normal NLRP3 inflammasome regulation. The absence of an increased volume of *hexb*^−/−^ BMMϕ lysosomes is likely a reflection of the relatively low throughput of substrates in the endo-lysosomal system of these hematopoietic cells maturing in the bone marrow. In contrast, mature RPMϕ accumulate substrates due to their scavenging/phagocytic activity that removes unwanted materials and cells from the peritoneum. It is relevant that the stages of disease that encompass NLRP3 dysregulation coincide with the period when IL-1β levels in the brain are significantly elevated [10]. We hypothesize that the loss of canonical regulation of NLRP3 inflammasome activity only in the symptomatic phases of disease results either directly or indirectly from substrate accumulation, reaching a critical threshold level and triggering uncontrolled inflammation. At this time, we have not identified the mechanism by which sphingolipid storage affects inflammasome function, but this clearly merits future studies.

The ingestion of crystalline material causes lysosome swelling, rupture, and leakage of contents into the cytosol [23,37,38]. Silica-mediated IL-1β production is partially dependent upon cathepsin B activity [23]. However, we were unable to measure any change in the amount of IL-1β released when *hexb*^−/−^ Mϕ were primed by LPS in the presence of the cathepsin B inhibitor CA-074-Me. Lysosomal leakage, rather than loss of integrity, has been implicated in several LSDs [39] but is typically associated with lysosome overloading via the uptake of neuronal material such as myelin, which leads to cathepsin-dependent pathology [40]. It may be relevant that heat-shock-protein-70-mediated lysosomal stabilization in murine sphingolipidoses models, including SD, provides significant therapeutic benefit [41], but effects on induction of inflammation were not reported.

Published studies of SD mice have shown that increased levels of IL-1β in the CNS are concurrent with the onset of chronic neuroinflammation [10] and that beneficial therapeutic interventions via substrate reduction therapy, bone marrow transplantation, or neural stem cell transplantation reduce levels of IL-1β [10,30]. We therefore examined whether selective ablation of IL-1β activity might modulate the course of the disease and used two experimental approaches to test this hypothesis: genetic deletion of *Il1r1* in *hexb*^−/−^ mice and competitive inhibition of IL-1β activity with recombinant IL-1Ra. We found a modest but statistically significant increase in the lifespan of *hexb*^−/−^ mice with both genetic deletion and blockade of IL-1β activity and a reduction in tremor amplitude consistent with an improvement in neurological function using the two approaches, reinforcing the potential for IL-1β to be a candidate therapeutic target for disease-modifying treatment. The extension in survival was similar to that reported previously for the treatment of SD mice with a selection of non-steroidal anti-inflammatory drugs [11]. A comparable investigation of the contribution of TNFα to disease progression in SD mice in which cytokine activity was deleted genetically through the creation of *hexb*^−/−^*tnfa*^−/−^ mice increased lifespan by approximately 15% as compared with *hexb*^−/−^ animals, together with a reduction in astrogliosis and neuronal apoptosis but an unchanged profile of microglia activation [26]. These data point to greater benefit from neutralization of TNFα in comparison with that achieved by blockade of IL-1β. This could indicate that the two inflammatory cytokines have distinct contributions to the disease process and potentially temporal differences in their involvement during disease progression. The study by Abo-Ouf and colleagues [26] also confirmed the absence of increased levels of IL-6 in *hexb*^−/−^ mice as compared with controls, suggesting that the cytokine does not contribute significantly to disease.

There is a good deal of evidence linking inflammasomes, including NLRP3, with the induction of neuroinflammation and promotion of neurodegeneration, encompassing disorders such as Alzheimer’s disease, Parkinson’s disease, and amyotrophic lateral sclerosis [42]. It is argued that the aberrant assembly and activation of inflammasome complexes and, as a result, the generation of IL-1β, can amplify the pathological changes in neurodegeneration [43], and we would suggest that our data are indicative of comparable pathways in SD. Although mechanistically this has been explored to a much lesser extent in LSDs, in Gaucher disease, the accumulation of glucosylceramide causes inhibition of autophagy and subsequent inflammasome activation and generation of IL-1β [44].

Significantly, we also obtained a degree of therapeutic benefit in murine SD by pharmacological inhibition of IL-1β activity by injection of anakinra. This receptor antagonist has been used to successfully target the cytokine not only in autoinflammatory diseases [45] but also in other conditions affecting the skeleton (such as rheumatoid arthritis) and systemic (Behcet’s disease) and common (gout) inflammatory disorders [24]. Potentially, this strategy could be transferred to the clinical setting for SD. Other specific therapeutics in clinical use that block IL-1 exist that could be trialed in patients and include soluble decoy receptor (rilonacept) and neutralizing antibodies (such as canakinumab), but these are ineffective in mice due to poor species cross-reactivity and were not evaluated in this study [24]. *hexb*^−/−^ mice lack all enzyme activity and show rapid disease onset and decline. In contrast, the vast majority of patients have some residual enzyme activity, albeit at variable levels according to their specific mutation(s) and, as consequence, have a heterogeneous clinical course [1]. It would therefore be anticipated that targeting IL-1β will provide greater benefit in patients with slower rates of disease progression. Finally, combination therapy, involving antagonism of IL-1 together with approaches that block other pro-inflammatory cytokines such as TNFα [46], may provide additive or synergistic benefits in SD patients and merit future evaluation.

In summary, we describe for the first time dysregulation of the NLRP3 inflammasome in murine SD and demonstrate that IL-1β actively contributes to disease pathogenesis, suggesting that this pro-inflammatory cytokine may be a target for clinical intervention as a potential disease-modifying therapy in this currently intractable LSD.

## 5. Limitations of Our Study

The major objective of this study was to ascertain whether IL-1β is pathogenic in SD and obtain evidence that it may be a target for beneficial therapy in the disease. Although we show the aberrant generation of the cytokine, we have not determined the molecular mechanism responsible for overriding the requirement for the two-step regulation of IL-1β production. Furthermore, we were only able to perform a pilot study of the benefit of anakinra administration in *hexb*^−/−^ mice because of limited availability and therefore were unable to measure multiple parameters to more precisely evaluate effectiveness, nor measure the extent of neutralization of cytokine activity.

## Figures and Tables

**Figure 1 cells-14-00035-f001:**
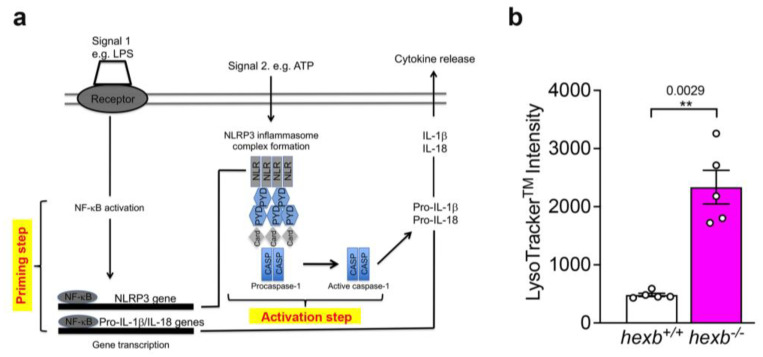
*hexb*^−/−^ resident peritoneal macrophages (RPMϕ) isolated from 14-week-old mice display significantly greater LysoTracker^TM^ staining intensity in comparison with age-matched *hexb*^+/+^ RPMϕ. Panel (**a**), cartoon of the 2-signal regulation of IL-1β production by NLRP3 inflammasome. Priming and activation steps are highlighted in yellow. (**b**) Histogram of relative intensity of LysoTracker^TM^ staining of *hexb*^+/+^ RPMϕ (open columns) and *hexb*^−/−^ RPMϕ (magenta-filled columns). Values, mean ± SEM, n = 5. Statistical analysis, Student’s *t*-test. ** *p* < 0.0029. Data are representative of a minimum of 3 independent experiments.

**Figure 2 cells-14-00035-f002:**
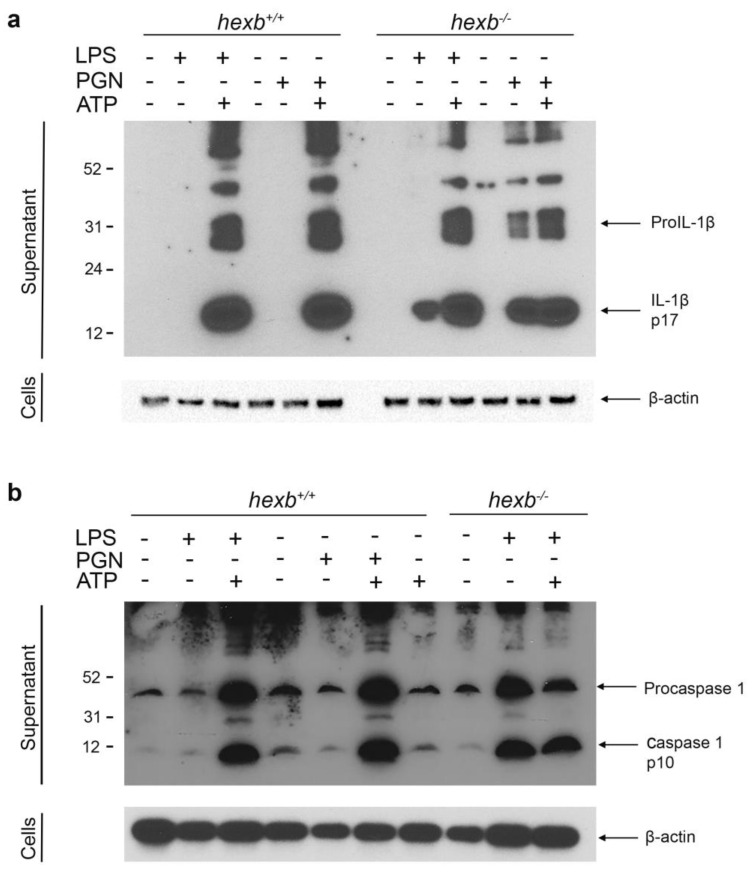
*hexb*^−/−^ resident peritoneal macrophages (RPMϕ) but not *hexb*^+/+^ RPMϕ isolated from 14-week-old mice secrete significant quantities of IL-1β and caspase-1 in response to priming of NLRP3 inflammasome. (**a**) Upper panel: Western blot of culture supernatants of *hexb*^+/+^ and *hexb*^−/−^ RPMϕ primed with either LPS or PGN or primed and activated with ATP and probed with anti-IL-1β antisera. Lower panel: Western blot of cell lysates of *hexb*^+/+^ and *hexb*^−/−^ RPMϕ primed with either LPS or PGN or primed and activated with ATP and probed with anti-β-actin antisera. (**b**) Upper panel: Western blot of culture supernatants of *hexb*^+/+^ and *hexb*^−/−^ RPMϕ primed with either LPS or PGN or primed and activated with ATP and probed with anti-caspase-1 antisera. Lower panel. Western blot of cell lysates of *hexb*^+/+^ and *hexb*^−/−^ RPMϕ primed with either LPS or PGN or primed and activated with ATP and probed with anti-β-actin antisera. Migration of ProIL-1β and mature IL-1β or Procaspase-1 and caspase-1 are indicated with arrows. Migration of molecular weight markers as indicated. Data are representative of two independent experiments.

**Figure 3 cells-14-00035-f003:**
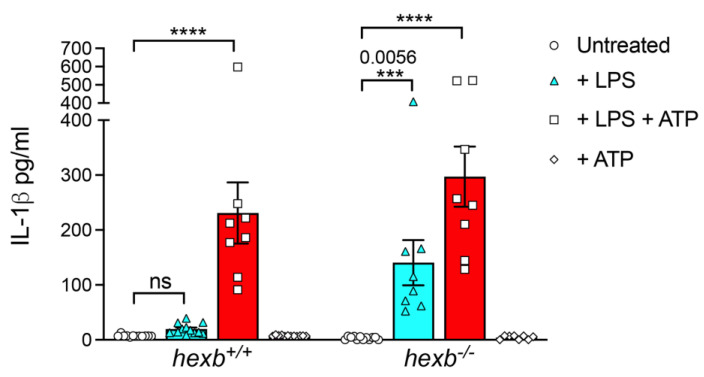
*hexb*^−/−^ resident peritoneal macrophages (RPMϕ) from 14-week-old mice but not *hexb*^+/+^ RPMϕ secrete significant levels of IL-1β following LPS priming. Histogram of cytokine concentrations (pg/mL) determined by specific ELISA in culture supernatants of *hexb*^−/−^ and *hexb*^+/+^ RPMϕ either untreated (unfilled circles), primed with LPS (cyan filled columns), primed with LPS and activated with ATP (red filled columns) or activated only (unfilled diamonds). Data are mean ± SEM, n = 8. Statistical analysis, One-way ANOVA. **** *p* < 0.0001 or *** *p* = 0.0056. Results are representative of 2 independent experiments.

**Figure 4 cells-14-00035-f004:**
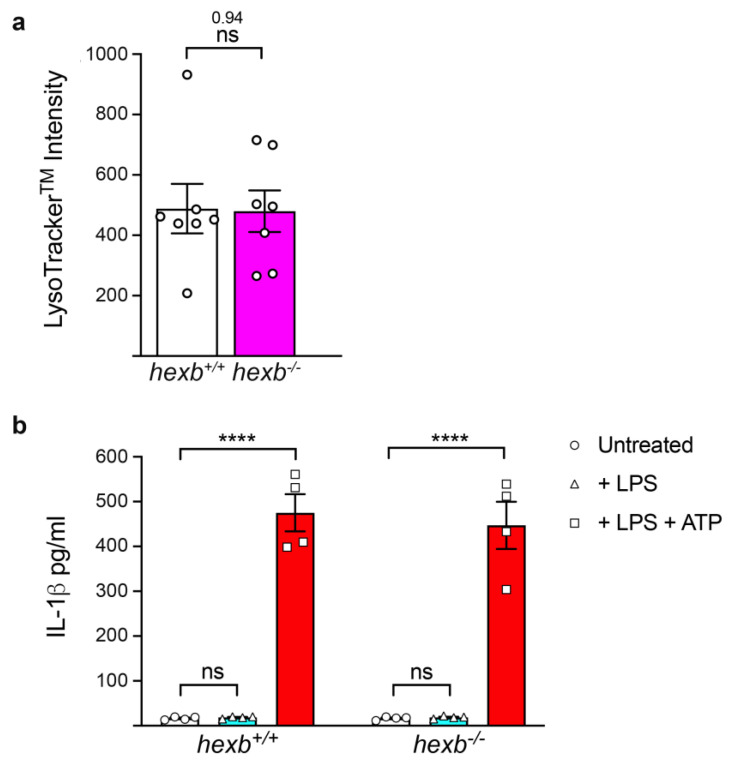
BMMϕ derived from 14-week-old *hexb*^−/−^ mice do not display enhanced LysoTracker^TM^ staining or aberrant production of IL-1β. (**a**) Histogram of relative intensity of LysoTracker^TM^ staining of *hexb*^+/+^ BMMϕ (open columns) and *hexb*^−/−^ BMMϕ (magenta-filled columns). Values, mean ± SEM, n = 7. Statistical analysis, Student’s *t*-test. ns, not significant. Data are representative of a minimum of 3 independent experiments. (**b**) Histogram of ELISA measurements of IL-1β concentrations in culture supernatants of BMMϕ derived from 14-week-old *hexb*^+/+^ and *hexb*^−/−^ mice, either untreated (open columns), LPS treated (cyan filled columns) or LPS + ATP (red filled columns). Values, mean ± SEM, n = 4. Statistical analysis, one-way ANOVA. **** *p* < 0.0001, ns, not significant. Data are representative of a minimum of 3 independent experiments.

**Figure 5 cells-14-00035-f005:**
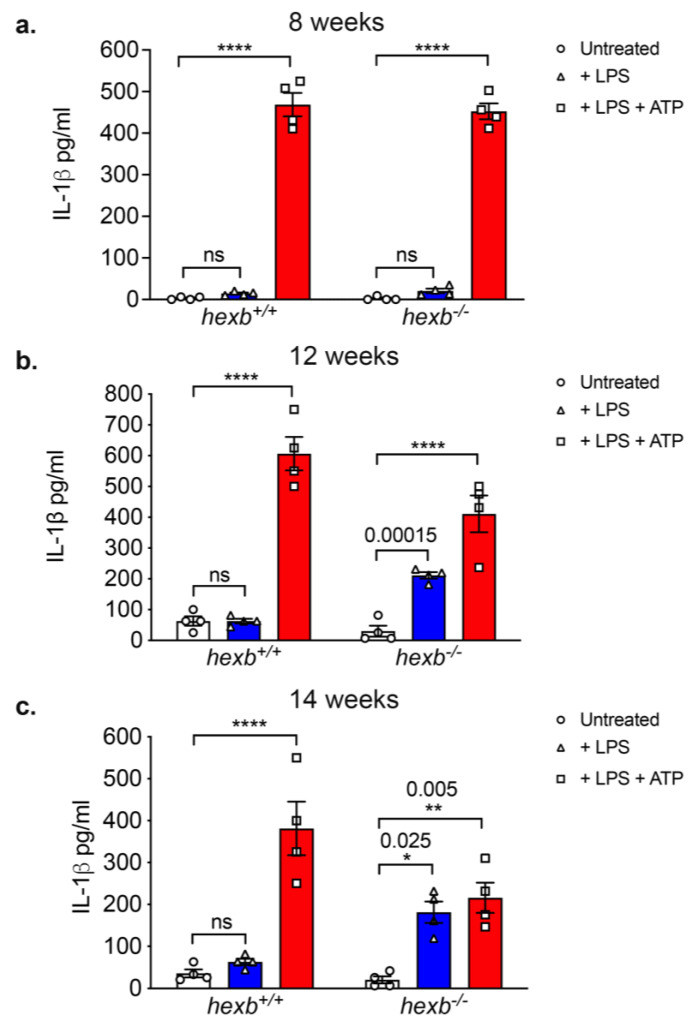
*hexb*^−/−^ RPMϕ isolated from 12-week-old and 14-week-old symptomatic but not 8-week-old non-symptomatic mice display aberrant IL-1β production after priming. Histograms of ELISA measurements of IL-1β concentrations in supernatants of RPMϕ isolated from 8-week-old (**a**), 12-week-old (**b**), and 14-week-old (**c**) *hexb*^+/+^ and *hexb*^−/−^ mice either untreated (open columns), primed with LPS (blue columns), or primed with LPS and activated with ATP (red columns). Data, mean ± SEM. n = 5 replicates for each sample. Statistical analysis, Student’s *t*-test **** *p* < 0.0001, ** *p* < 0.0.01, * *p* < 0.05. ns, not significant. Data are representative of three independent experiments.

**Figure 6 cells-14-00035-f006:**
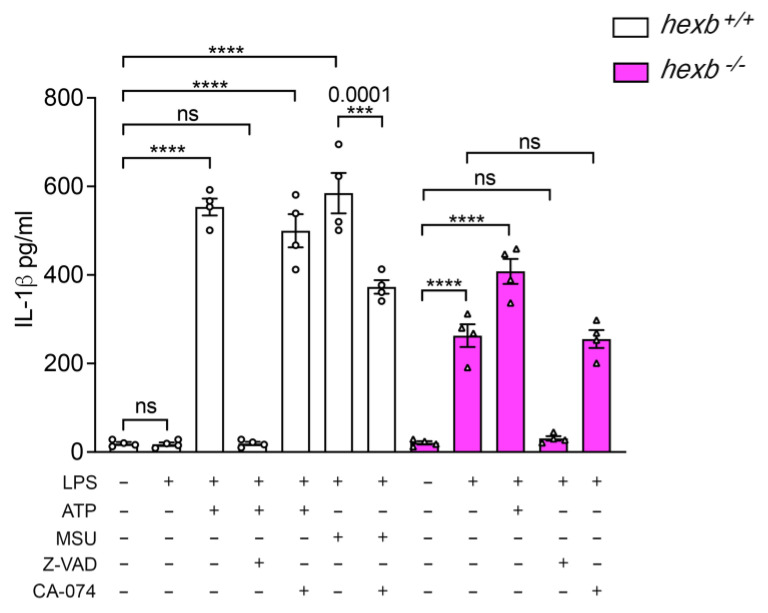
Inhibition of caspase-1 but not cathepsin B activity significantly reduces IL-1β production by LPS-primed *hexb*^−/−^ RPMϕ isolated from 14-week-old mice. Histogram of ELISA determinations of IL-1β concentrations in supernatants of RPMϕ from 14-week-old *hexb*^+/+^ (open columns) or *hexb*^−/−^ (magenta columns) mice either untreated, primed with LPS, primed with LPS and activated with ATP, or primed with LPS and incubated with monosodium urate crystals (MSU) in the absence or presence of the caspase-1 inhibitor zVAD-fmk or cathepsin B inhibitor, CA-074. Data are mean ± SEM, n = 5 replicates per treatment. Statistical analysis, one-way ANOVA. **** *p* < 0.0001, *** *p* = 0.001, ns, not significant. Data are representative of three independent experiments.

**Figure 7 cells-14-00035-f007:**
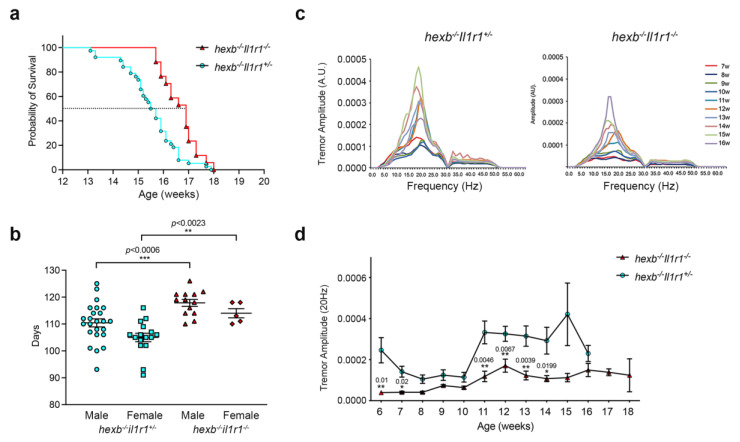
*hexb*^−/−^*Il1r1*^−/−^ *mice* have a significantly extended lifespan and display improved tremors: (**a**) Kaplan–Meier survival plot of *hexb*^−/−^*Il1r1*^−/−^ mice as compared with *hexb*^−/−^*Il1r1*^+/*−*^ animals. (**b**) Both male and female *hexb*^−/−^*Il1r1*^−/−^mice have significantly extended lifespans in comparison with male and female *hexb*^−/−^*Il1r1*^+/*−*^ animals. Data are mean ± SEM, n = 5–24. Statistical analysis, Student’s *t*-test; *** *p* = 0.0006, ** *p* = 0.0023. (**c**) Profile of tremor amplitudes at multiple frequencies for *hexb*^−/−^*Il1r1*^+/−^ mice (**left**)) and *hexb*^−/−^*Il1r1*^−/−^ mice (**right**) at different ages. (**d**) Tremor amplitude at 20 Hz for *hexb*^−/−^*Il1r1*^+/−^ and *hexb*^−/−^*Il1r1*^−/−^ mice at different ages. Data shown are mean ± SEM, n = 5–20. Statistical analysis, two-way ANOVA. Statistical significances are as indicated. Data are representative of two independent experiments.

**Figure 8 cells-14-00035-f008:**
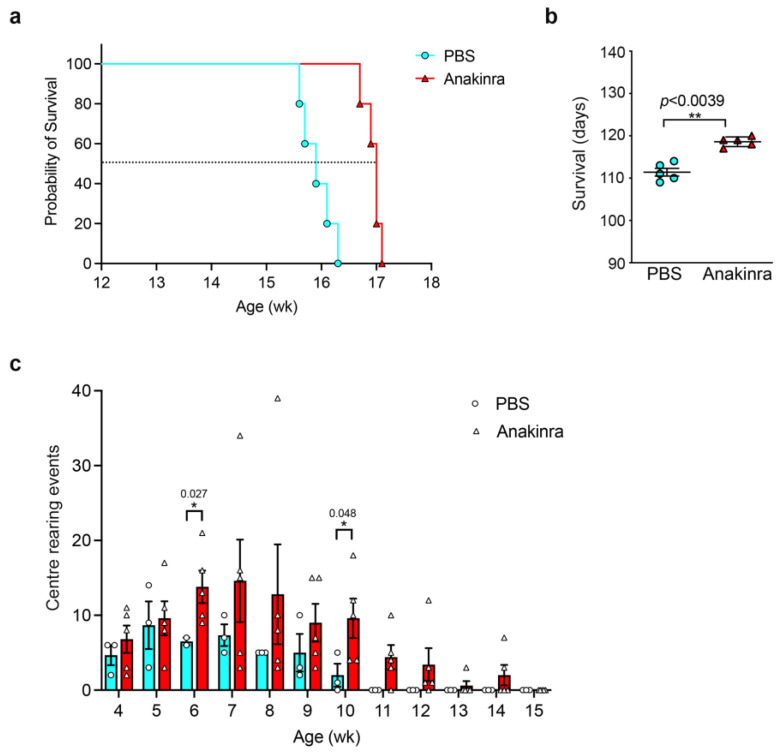
Blockade of IL-1β activity with anakinra significantly extends lifespan of *hexb*^−/−^ mice and improves neurological function: (**a**) Kaplan–Meier survival plot of *hexb*^−/−^ mice administered with anakinra (red columns, open triangles) or vehicle (cyan columns, open circles). (**b**) *hexb*^−/−^ mice treated with anakinra (red triangles) have a significantly increased lifespan in comparison with vehicle-treated mice (cyan circles). Data are mean ± SEM, n = 5. ** *p* = 0.0039. Student’s *t*-test. (**c**) Frequency of center-rearing events by anakinra-treated *hexb*^−/−^ mice (red columns) and vehicle-treated (cyan columns) animals. Data are mean± SEM, n = 3–5. Statistical analysis, Student’s *t*-test; statistical significance values are as indicated. Data are representative of 2 independent experiments.

**Table 1 cells-14-00035-t001:** Summary of IL-1β secretion by *hexb*^+/+^ or *hexb*^−/−^ resident peritoneal macrophages isolated from mice of different ages and then primed with LPS or primed with LPS and activated with ATP.

Enotype	Age (Weeks)	Priming Only	Priming + Activation
*hexb* ^+/+^	8	🗴	✓
	12	🗴	✓
	14	🗴	✓
*hexb* ^−/−^	8	🗴	✓
	12	✓	✓
	14	✓	✓

🗴 Insignificant IL-1β secretion. ✓ Significant IL-1β secretion.

## Data Availability

All data are available on request. Correspondence and requests for materials should be addressed to N.P. and Frances M. Platt.

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
