# Peer review of "Dysregulation of the NLRP3 Inflammasome and Promotion of Disease by IL-1β in a Murine Model of Sandhoff Disease"

_cells, 2025, doi:10.3390/cells14010035_

Round 1
Reviewer 1 Report
Comments and Suggestions for Authors
The authors describe NLRP3 inflammasome, a coordinator of pro-inflammatory responses that is, under commonly understood mechanisms, primed and activated by lipopolysaccharide and ATP, which in turn leads to IL-1 secretion.
The authors found, however, that in symptomatic SD mice, no ATP is required for the priming of NLPRP3, suggesting dysregulation in SD symptomatic mice is related to glycosphingolipid storage. This dysregulation appears to be dependent on caspase activity but not on cathepsin B. This finding was unexpected.
The authors tested these concepts in 2 mouse model experiments:
1) A hexb-/-Il1r1-/- double knockout mouse
2) A hexb-/- mouse treated with anakinra, an IL-1 receptor antagonist.
3) In both the hexb-/-Il1r1-/- double knockout mouse and the bexn-/- mouse treated with anakinra, there was observed a significant extension in lifespan and improvement in neurological function.
The authors evaluated ability to stimulate IL-1beta secretion in peritoneal cells with the 1 step process compared to the 2 step process in hexb-/- and heb+/+ mice, at 8 weeks (presymptomatic by behavioral tests in hexb-/-mice), 12 weeks and 14 weeks (the latter 2 in which hexb-/- mice are symptomatic). IL-1 beta was only able to be stimulated with the 2-step process in “symptomatic” mice.
The authors observed that dysregulation of NLPRP3 in symptomatic SD mice is dependent on caspases, but not on cathepsin B activity, whereas in hexb+/+mice both capspases and cathepsin B play a role in NLPRP3 regulation in terms of IL-1beta production.
A mouse model of hexb-/- mouse model that also lacks interleukin 1 receptor 1, showed improved lifespan and decreased tremor compared to the bexb-/- mice that have interleukin 1 receptor 1. Consistent with this finding, inhibition with an interleukin 1 receptor antagonist, anakinra, showed improved lifespan and rearing activity (ability to stand on hind legs) in the hexb-/- mouse.
The authors conclude that “combination therapy, involving antagonism of IL-1 together with approaches that block other pro-inflammatory cytokines such as TNFa [49] may provide additive or synergistic benefit in 492 SD patients and merit future evaluation.”
This research project identifies critical inflammatory processes involved the central nervous system pathology of GM2-gangliosidoses. We have known for many years that GM2-gangliosidis involves increased level of mRNA expression in central nervous system tissue for numerous inflammatory mediators. It was also recognized that IL-1 beta and TNF seemed and play a prominent role in CNS pathology of the GM2-gangliosidoses, but whether or not these inflammatory markers are worthwhile targeting with therapy has not been demonstrated. Importantly, this study is able to demonstrate a mechanism RP3 dysregulation as a key process contributing to pathology and disease progression in the GM2-gangliosidoses, and also demonstrates potential benefits of therapies targeting IL-1 beta in the GM2-gangliosidoses. The concept of combination of dual therapy targeting both IL-1 beta and TNF, is also presented as a potentially beneficial approach, worthy of future research.
Recommendation:
1) “Anakinra:” should not be capitalized, unless it is the first word in a sentence.
Author Response
Reviewer 1 just wanted Anakinra decapitalising which we have done.
Reviewer 2 Report
Comments and Suggestions for Authors
The article “Dysregulation of the NLRP3 inflammasome and promotion of disease by IL-1β in a murine model of Sandhoff Disease” emphasises the role of cytokine IL-1β on the landscape of neurodegenerative lysosomal storage diseases and demonstrates how NLRP3 inflammasome dysfunction regulate active IL-1β secretion and disease progression in a murine disease model (hexb-/-). Although the article successfully expands our understanding of the subject matter, some concerns need to be addressed before the manuscript is fit for publication –
1. The presence of multiple protein bands in the western blot is confusing. Expression of a housekeeping gene (GAPDH or β-actin) should be added for comparative analysis.
2. The expression profile of other pro-inflammatory cytokines like IL-6 and IL-18 should also be given to highlight the impact of IL-1β in the disease model.
3. Inhibition of caspase1, cathepsin B and IL-1R by their respective inhibitors should be validated by western blot and/or qPCR.
4. The focal point of this article, i.e. the influence of IL-1β in neurodegenerative LSDs and how disease progression occurs in hexb-/- murine disease model by bypassing the conventional cathepsin B activation of the inflammasome via LPS priming and NF-Ò¡β/NLRP3/caspase1/IL-1β axis should be featured in the title and discussion part of the article.

Author Response
Comments and Suggestions for Authors from Reviewer 2
The article “Dysregulation of the NLRP3 inflammasome and promotion of disease by IL-1β in a murine model of Sandhoff Disease” emphasises the role of cytokine IL-1β on the landscape of neurodegenerative lysosomal storage diseases and demonstrates how NLRP3 inflammasome dysfunction regulate active IL-1β secretion and disease progression in a murine disease model (hexb-/-). Although the article successfully expands our understanding of the subject matter, some concerns need to be addressed before the manuscript is fit for publication –
- The presence of multiple protein bands in the western blot is confusing. Expression of a housekeeping gene (GAPDH or β-actin) should be added for comparative analysis.
This was an oversight on our part. We have now included blots of b-actin expression to show equivalent protein amounts. The antibodies used to detect caspase-1 and IL-1b are also reactive with ProIL-1b and Procaspase-1. We have indicated on the blots the bands that represent the pro-forms of the two proteins.
- The expression profile of other pro-inflammatory cytokines like IL-6 and IL-18 should also be given to highlight the impact of IL-1β in the disease model.
We have referenced a publication that reported the absence of a significant increase in the levels of IL-6 in SD, strongly suggesting that this cytokine is unlikely to have a significant impact upon disease.
- Inhibition of caspase1, cathepsin B and IL-1R by their respective inhibitors should be validated by western blot and/or qPCR.
In Fig 6. we include ELISA data that confirmed the activities of the inhibitors a) z-vad, a caspase-1 inhibitor, as expected significantly reduced the secretion of IL-1b following priming and activation of hexb+/+ RPMF and b) that CA-074, an inhibitor of cathepsin B as expected significantly reduced the secretion of IL-1b following priming of hexb+/+ RPMF with LPS and lysosome rupture by ingestion of MSU particles.
Mechanistically, these compounds do not i either inhibit IL-1b transcription or protein expression. The positive controls included in Fig 6, do demonstrate biological inhibition. The studies suggested by the reviewer would not therefore confirm inhibition.
The same argument applies to the suggestion to validate inhibition by IL-1RA by WB or qPCR.
- The focal point of this article, i.e. the influence of IL-1β in neurodegenerative LSDs and how disease progression occurs in hexb-/- murine disease model by bypassing the conventional cathepsin B activation of the inflammasome via LPS priming and NF-Ò¡β/NLRP3/caspase1/IL-1β axis should be featured in the title and discussion part of the article.
We have taken on board the suggestion of the reviewer and have included a paragraph within the discussion that includes references to the involvement of the cytokine IL-1b in neuroinflammation and neurodegeneration more generally. We also refer to evidence that aberrant assembly and activation of inflammasomes is also associated with the pathology of neurodegenerative disorders, including another LSD.
We feel the title accurately reflects the contents of our manuscript. We are not in this study generalizing to other LSDs because we can only comment on the one that has been studied. Indeed, as we indicate in the discussion although inflammation is a shared characteristic of LSDs, there is published evidence that the underlying causes are disease-specific. Indeed, this also includes inflammasomes.
Reviewer 3 Report
Comments and Suggestions for Authors
Lysosomal dysfunction is associated with NLRP3 inflammasome activation. In lysosomal disorders, such as Sandhoff disease (SD), there is upregulation of both non-canonical and canonical inflammasome genes (PMID: 36519759). In this manuscript, the authors isolated resident peritoneal macrophages (RPM) and bone marrow-derived macrophages (BMDM) from Sandhoff disease mice (hexb-/-). They found that LPS/ATP-induced IL-1β release and caspase-1 activation were not significantly different between wild-type and hexb-/- mice. However, RPM from symptomatic SD mice (12 and 14 weeks old) displayed higher IL-1β release upon LPS treatment compared to wild-type mice. Interestingly, BMDM did not exhibit aberrant IL-1β production. The authors also generated hexb-/- Il1r1-/- mice and demonstrated that IL1R ablation or Anakinra treatment significantly prolonged lifespan and reduced tremors in these mice. While the study largely makes sense, several flaws need to be addressed:
Major Concerns:
1. The manuscript does not provide direct evidence supporting NLRP3 inflammasome activation in the cell models or in vivo tissues. Critical experiments such as demonstrating NLRP3 and ASC oligomerization, as well as results from NLRP3-selective inhibitors like MCC950 or CY-09, are missing.
2. The authors should provide immunoblot results for IL-1β, caspase-1, and NLRP3 in BMDM to clarify whether lysosomal dysfunction impacts these pathways in this cell type.
3. Unlike BMDM, RPM are isolated from peritoneal lavage without enrichment or purification, potentially leading to contamination with other immune cells. This issue has not been discussed or addressed and may confound the results.
4. The manuscript lacks clarity on the interpretation of LysoTracker™ staining. Does higher LysoTracker™ density indicate increased lysosomal damage? Why do hexb-/- BMDM not exhibit elevated lysosomal damage compared to RPM? More precise experiments are needed to compare and validate lysosomal damage in RPM and BMDM.
5. Previous studies suggest that peptidoglycan not only primes NLRP3 signaling but can also directly activate NLRP3 (PMCID: PMC5534359). The blot for caspase-1 in the PGN group for hexb-/- in Figure 2B is missing.
6. Data on tremor measurements are missing in Anakinra treatment.
7. There are errors in Figure 5B labeling.
Author Response
Lysosomal dysfunction is associated with NLRP3 inflammasome activation. In lysosomal disorders, such as Sandhoff disease (SD), there is upregulation of both non-canonical and canonical inflammasome genes (PMID: 36519759). In this manuscript, the authors isolated resident peritoneal macrophages (RPM) and bone marrow-derived macrophages (BMDM) from Sandhoff disease mice (hexb-/-). They found that LPS/ATP-induced IL-1β release and caspase-1 activation were not significantly different between wild-type and hexb-/- mice. However, RPM from symptomatic SD mice (12 and 14 weeks old) displayed higher IL-1β release upon LPS treatment compared to wild-type mice. Interestingly, BMDM did not exhibit aberrant IL-1β production. The authors also generated hexb-/- Il1r1-/- mice and demonstrated that IL1R ablation or Anakinra treatment significantly prolonged lifespan and reduced tremors in these mice. While the study largely makes sense, several flaws need to be addressed:
Major Concerns:
- The manuscript does not provide direct evidence supporting NLRP3 inflammasome activation in the cell models or in vivo tissues. Critical experiments such as demonstrating NLRP3 and ASC oligomerization, as well as results from NLRP3-selective inhibitors like MCC950 or CY-09, are missing.
Response: We acknowledge the point raised by the reviewer. However, the objective of our study was primarily to obtain evidence for a pathogenic role of IL-1b in GM2 gangliosidosis and its potential as a therapeutic target. We have now included a statement of the limitations of our study and questions that have yet to be addressed which includes detailed studies of inflammasome activation and the molecular basis for inflammasome dysregulation.
- The authors should provide immunoblot results for IL-1β, caspase-1, and NLRP3 in BMDM to clarify whether lysosomal dysfunction impacts these pathways in this cell type.
Response: Fig 4 shows that production of IL-1b via priming and activation of NLRP3 inflammasome is not affected in hexb-/- BMMF and the response is comparable to that of hexb+/+ BMMF (i.e. there is no evidence of dysregulation of the inflammasome in the former). Therefore, the argument is that induction of caspase-1, NLRP3 and IL-1b in hexb-/- BMMF will likely not be changed.
- Unlike BMDM, RPM are isolated from peritoneal lavage without enrichment or purification, potentially leading to contamination with other immune cells. This issue has not been discussed or addressed and may confound the results.
Response: Although this scenario is unlikely to explain the dysregulation production of IL-1b by hexb-/- RPMF as we select out adherent cells that greatly enriches RPMF we have now included a statement to address this point in the discussion.
- The manuscript lacks clarity on the interpretation of LysoTracker™ staining. Does higher LysoTracker™ density indicate increased lysosomal damage? Why do hexb-/- BMDM not exhibit elevated lysosomal damage compared to RPM? More precise experiments are needed to compare and validate lysosomal damage in RPM and BMDM.
Response: The reviewer is perhaps mistaken as to the interpretation of greater LysoTracker staining intensity. Increased LysoTracker staining is not an indication or measure of lysosome damage, but that of the relative size of the total acidic compartment of the cell. A shared feature of lysosomal storage diseases in that the size or number of lysosomes or a combination of the two is greater than that of healthy or non-diseased cells (te Vruchte et al 2014). Enhanced LysoTracker is therefore a measure of the extent of substrate storage in LSDs. The level of substrate storage, as we explain in the discussion, is dependent upon the rate of throughput of substrates in the endo-lysosomal system and whilst this is relatively high for resident peritoneal macrophages because of their scavenging activities, it is low for BMMF because of their relative inactivity during their in vitro differentiation from precursors.
- Previous studies suggest that peptidoglycan not only primes NLRP3 signaling but can also directly activate NLRP3 (PMCID: PMC5534359). The blot for caspase-1 in the PGN group for hexb-/- in Figure 2B is missing.
Response: We thank the reviewer for bringing this reference to our attention. However, after reading the paper we feel this mechanism is very unlikely to be responsible for our results. The study by Wolf et al is focused upon the effects of PGN on BMMF that have been primed with LPS prior to exposure to PNG. Our manuscript includes data confirming that hexb-/- BMMF do not display inflammasome dysregulation and that the effects of priming of hexb+/+ RPMF with PGN and then activation with ATP are similarly not different.
In Fig 2 we show that the aberrant production of IL-1b occurs when hexb-/- RPMF are primed with either LPS or with PGN (i.e. is independent of the nature of the priming stimulus). Having confirmed this, we therefore demonstrated aberrant caspase-1 activation in hexb-/- RPMF after priming with LPS only.
- Data on tremor measurements are missing in Anakinra treatment.
Response: We indicate that unfortunately we were only able to undertake a pilot study of the benefit of anakinra in hexb-/- mice because of the limited amount of drug availability to us. We were therefore not able to measure multiple parameters in treated animals.
- There are errors in Figure 5B labeling.
Response: In error we uploaded the wrong version of this figure. We have now replaced it with the correct version.
Round 2
Reviewer 2 Report
Comments and Suggestions for Authors
The authors addressed all the concerns regarding the article. The article is fit for publication.
Reviewer 3 Report
Comments and Suggestions for Authors
No more comments